# CoPriv: Network/Protocol Co-Optimization for Communication-Efficient Private Inference

**Wenxuan Zeng**
Peking University
zwx.andy@stu.pku.edu.cn

**Meng Li**[*]
Peking University
meng.li@pku.edu.cn

**Haichuan Yang**
Meta AI
haichuan@meta.com

**Wen-jie Lu**
Ant Group
juhou.lwj@antgroup.com

**Runsheng Wang**
Peking University
r.wang@pku.edu.cn

**Ru Huang**
Peking University
ruhuang@pku.edu.cn

## Abstract

Deep neural network (DNN) inference based on secure 2-party computation (2PC) can offer cryptographically-secure privacy protection but suffers from orders of magnitude latency overhead due to enormous communication. Previous works heavily rely on a proxy metric of ReLU counts to approximate the communication overhead and focus on reducing the ReLUs to improve the communication efficiency. However, we observe these works achieve limited communication reduction for state-of-the-art (SOTA) 2PC protocols due to the ignorance of other linear and non-linear operations, which now contribute to the majority of communication. In this work, we present CoPriv, a framework that jointly optimizes the 2PC inference protocol and the DNN architecture. CoPriv features a new 2PC protocol for convolution based on Winograd transformation and develops DNN-aware optimization to significantly reduce the inference communication. CoPriv further develops a 2PC-aware network optimization algorithm that is compatible with the proposed protocol and simultaneously reduces the communication for all the linear and non-linear operations. We compare CoPriv with the SOTA 2PC protocol, CrypTFlow2, and demonstrate $2.1\times$ communication reduction for both ResNet-18 and ResNet-32 on CIFAR-100. We also compare CoPriv with SOTA network optimization methods, including SNL, MetaPruning, etc. CoPriv achieves $9.98\times$ and $3.88\times$ online and total communication reduction with a higher accuracy compared to SNL, respectively. CoPriv also achieves $3.87\times$ online communication reduction with more than 3% higher accuracy compared to MetaPruning.

## 1 Introduction

Deep learning has been applied to increasingly sensitive and private data and tasks, for which privacy emerges as one of the major concerns. To alleviate the privacy concerns when deploying the deep neural network (DNN) models, secure two-party computation (2PC) based DNN inference is proposed and enables cryptographically-strong privacy guarantee [39, 28, 48, 44, 25, 24, 47].

Secure 2PC helps solve the following dilemma [28, 48, 44]: the server owns a private model and the client owns private data. The server is willing to provide the machine learning as a service (MLaaS) but does not want to give it out directly. The client wants to apply the model on the private data without revealing it as well. Secure 2PC frameworks can fulfill both parties' requirements: two parties can learn the inference results but nothing else beyond what can be derived from the results.

---

[*]Corresponding author.

37th Conference on Neural Information Processing Systems (NeurIPS 2023).

The privacy protection of secure 2PC-based inference is achieved at the cost of high communication complexity due to the massive interaction between the server and the client [31, 48, 43, 25]. This leads to orders of magnitude latency gap compared to the regular inference on plaintext [31, 43]. A 2PC-based inference usually has two stages, including a pre-processing stage that generates the input independent helper data and an online stage to process client's actual query [31, 48, 43, 25, 24]. Because the helper data is independent of client's query, previous works [48, 25] assume the pre-processing stage is offline and thus, focus on optimizing the communication of the input dependent stage. Specifically, [48, 19, 26, 6, 7, 32] observe ReLU accounts for significant communication and ReLU count is widely used as a proxy metric for the inference efficiency. Hence, the problem of improving inference efficiency is usually formulated as optimizing networks to have as few ReLUs as possible. For example, DeepReduce [26] and SNL [7] achieve more than $4\times$ and $16\times$ ReLU reduction with less than 5% accuracy degradation on the CIFAR-10 dataset.

However, such assumption may no longer be valid for state-of-the-art (SOTA) 2PC protocols. We profile the communication of ResNet-18 with different ReLU optimization algorithms, including DeepReduce [26], SENet [32], and SNL [7] and observe very limited communication reduction as shown in Figure 1(b). This is because, on one hand, the communication efficiency of ReLU has been drastically improved over the past few years from garble circuit (GC) to VOLE oblivious transfer (OT) as shown in Figure 1(a) [28, 43, 24], and ReLU only accounts for 40% and 1% of the online and total communication, respectively. On the other hand, recent studies suggest for MLaaS, it is more im-

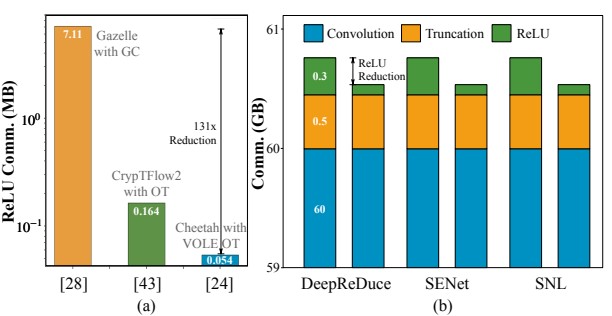

Figure 1: (a) Communication for ReLU communication has reduced by $131\times$ from [28] to [24]; (b) existing network optimizations suffers from limited reduction of online and total communication.

portant to consider inference request arrival rates rather than studying individual inference in isolation [18]. In this case, the pre-processing communication, which is significantly higher than the online communication as shown in Figure 1(b), *cannot be ignored and may often be incurred online as there may not be sufficient downtime for the server to hide its latency* [18].

Therefore, to improve the efficiency of 2PC-based inference, in this paper, we argue that both pre-processing and online communication are important and propose CoPriv to jointly optimize the 2PC protocol and the DNN architecture. CoPriv first optimizes the inference protocol for the widely used 3-by-3 convolutions based on the Winograd transformation and proposes a series of DNN-aware protocol optimization. The optimized protocol achieves more than $2.1\times$ communication reduction for ResNet models [20] without accuracy impact. For lightweight mobile networks, e.g., MobileNetV2 [46], we propose a differentiable ReLU pruning and re-parameterization algorithm in CoPriv. Different from existing methods [26, 7, 32], CoPriv optimizes both linear and non-linear operations to simultaneously improve both the pre-processing and online communication efficiency.

With extensive experiments, we demonstrate CoPriv drastically reduces the communication of 2PC-based inference. The proposed optimized protocol with Winograd transformation reduces the convolution communication by $2.1\times$ compared to prior-art CrypTFlow2 [43] for both the baseline ResNet models. With joint protocol and network co-optimization, CoPriv outperforms SOTA ReLU-optimized methods and pruning methods. Our study motivates the community to directly use the communication as the efficiency metric to guide the protocol and network optimization.

## 2 Preliminaries

### 2.1 Threat Model

CoPriv focuses on efficient private DNN inference involving two parites, i.e., Alice (server) and Bob (client). The server holds the model with private weights and the client holds private inputs. At the end of the protocol execution, the client learns the inference results without revealing any

Table 1: Comparison with prior-art methods.

| Method | Protocol Opt. | Network Opt. | | |
|---|---|---|---|---|
| | | Conv. | Truncation | ReLU |
| [43, 42, 39, 8, 38, 29, 41] | ✓ | ✗ | ✗ | ✗ |
| [32, 7, 19, 40, 37, 6, 27, 26] | ✗ | ✗ | ✗ | ✓ |
| [48, 25] | ✓ | ✗ | ✗ | ✓ |
| [17] | ✓ | ✓ | ✗ | ✗ |
| CoPriv (ours) | ✓ | ✓ | ✓ | ✓ |

Table 2: Notations used in the paper.

| Notation | Meaning |
|---|---|
| $H, W$ | Height and width of output feature map |
| $C, K$ | Input and output channel number |
| $A, B, G$ | Winograd transformation matrices |
| $m, r$ | Size of output tile and convolution filter |
| $T$ | Number of tiles per input channel |
| $\lambda$ | Security parameter that measures the attack hardness |

information to the server. Consistent with previous works [39, 48, 43, 42, 26, 7, 32], we adopt an *honest-but-curious* security model in which both parties follow the protocol but also try to learn more from the information than allowed. Meanwhile, following [39, 28, 48, 43, 42], we assume no trusted third party exists so that the helper data needs to be generated by the client and the server.

## 2.2 Arithmetic Secret Sharing

CoPriv leverages the 2PC framework based on arithmetic secret sharing (ArSS). Specifically, an $l$-bit value $x$ is shared additively in the integer ring $\mathbb{Z}_{2^l}$ as the sum of two values, e.g., $\langle x \rangle_s$ and $\langle x \rangle_c$. $x$ can be reconstructed as $\langle x \rangle_s + \langle x \rangle_c \bmod 2^l$. In the 2PC framework, $x$ is secretly shared with the server holding $\langle x \rangle_s$ and the client holding $\langle x \rangle_c$.

ArSS supports both addition and multiplication on the secret shares. Addition can be conducted locally while multiplication requires helper data, which are independent of the secret shares and is generated through communication [43, 42]. The communication cost of a single multiplication of secret shares is $O(l(\lambda + l))$. When $t$ multiplications share the same multiplier, instead of simply repeating the multiplication protocol for $t$ times, [8, 31, 43] proposes a batch optimization algorithm that only requires $O(l(\lambda + tl))$ communication, enabling much more efficient batched multiplications.

Most 2PC frameworks use fixed-point arithmetic. The multiplication of two fixed-point values of $l$-bit precision results in a fixed-point value of $2l$-bit precision. Hence, in order to perform subsequent arithmetics, a truncation is required to scale the value down to $l$-bit precision. Truncation usually executes after the convolutions and before the ReLUs. It requires complex communication and sometimes leads to even more online communication compared to ReLU as shown in Figure 1. We refer interested readers to [43] for more details.

## 2.3 Winograd Convolution

We first summarize all the notations used in the paper in Table 2. Then, consider a 1D convolution with $m$ outputs and filter size of $r$, denoted as $F(m, r)$. With regular convolution, $F(m, r)$ requires $mr$ multiplications between the input and the filter. With the Winograd algorithm, $F(m, r)$ can be computed differently. Consider the example of $F(2, 3)$ below:

$$X = \begin{bmatrix} x_0 & x_1 & x_2 & x_4 \end{bmatrix}^\top \quad W = \begin{bmatrix} w_0 & w_1 & w_2 \end{bmatrix}^\top \quad Y = \begin{bmatrix} y_0 & y_1 \end{bmatrix}^\top$$

$$\begin{bmatrix} x_0 & x_1 & x_2 \\ x_1 & x_2 & x_3 \end{bmatrix} \begin{bmatrix} w_0 \\ w_1 \\ w_2 \end{bmatrix} = \begin{bmatrix} m_0 + m_1 + m_2 \\ m_1 - m_2 - m_3 \end{bmatrix} = \begin{bmatrix} y_0 \\ y_1 \end{bmatrix}$$

where $m_0, m_1, m_2, m_3$ are computed as

$$m_0 = (x_0 - x_2)w_0 \quad m_1 = (x_1 + x_2)(w_0 + w_1 + w_2)/2$$
$$m_3 = (x_1 - x_3)w_2 \quad m_2 = (x_2 - x_1)(w_0 - w_1 + w_2)/2.$$

With the Winograd algorithm, the number of multiplications is reduced from 6 in the regular convolution to 4. Similar Winograd transformation can be applied to 2D convolutions by nesting 1D algorithm with itself [34]. The 2D Winograd transformation $F(m \times m, r \times r)$, where the output tile size is $m \times m$, the filter size is $r \times r$, and the input tile size is $n \times n$, where $n = m + r - 1$, can be formulated as follows,

$$Y = W \circledast X = A^\top [(GWG^\top) \odot (B^\top XB)]A, \tag{1}$$

where $\circledast$ denotes the regular convolution and $\odot$ denotes element-wise matrix multiplication (EWMM). $A$, $B$, and $G$ are transformation matrices that are independent of $W$ and $X$ and can be computed based on $m$ and $r$ [34, 1].

With the Winograd algorithm, the multiplication of such a 2D convolution can be reduced from $m^2r^2$ to $n^2$, i.e., $(m + r - 1)^2$ at the cost of performing more additions which can be computed locally in private inference scenario. While the reduction of multiplication increases with $m$, $m = 2$ and $m = 4$ are most widely used for the better inference precision [34, 49, 3, 15]. More details about Winograd convolution are available in Appendix D.

## 2.4 Related Works

To improve the efficiency of private inference, existing works can be categorized into three classes, including protocol optimization [8, 38, 43, 42, 29, 41, 39], network optimization [26, 7, 19, 40, 37, 6, 32, 27, 51, 35], and joint optimization [48, 25, 17]. In Table 1, we compare CoPriv with these works qualitatively and as can be observed, CoPriv leverages both protocol and network optimization and can simultaneously reduce the online and total communication induced by convolutions, truncations, and ReLUs through network optimization. We leave more detailed review of existing works to Appendix A and more detailed comparison in terms of their techniques to Appendix F.

## 3 Motivation

In this section, we analyze the origin of the limited communication reduction of existing ReLU-optimized networks and discuss our key observations that motivates CoPriv.

**Observation 1: the total communication is dominated by convolutions while the online communication cost of truncations and ReLUs are comparable.** As shown in Figure 1, the main operations that incurs high communication costs are convolutions, truncations, and ReLUs. With CrypTFlow2, over 95% communication is in the pre-processing stage generated by convolutions, while the truncations and ReLUs requires similar communication. This observation contradicts to the assumption of previous works [7, 6, 19, 32]: on one hand, ReLU no longer dominates the online communication and simply reducing ReLU counts leads to diminishing return for online communication reduction; on the other hand, pre-processing communication cannot be ignored as it not only consumes a lot of power in the server and client but also can slow down the online stage. As pointed out by a recent system analysis [18], for a pratical server that accepts inference request under certain arrival rates, the pre-processing stage is often incurred online as there is insufficient downtime to hide its latency. Therefore, we argue that both the total and the online communication are crucial and need to be reduced to enable more practical 2PC-based inference.

**Observation 2: the communication cost scaling of convolution, truncation and ReLU is different with the network dimensions.** The communication cost of different operations scales differently with the network dimensions. For a convolution, the major operations are multiplications and additions. While additions are computed fully locally, multiplications requires extra communication to generate the helper data. Hence, the communication cost for a convolution scales linearly with the number of multiplications and the complexity is $O(CKr^2(\lambda + HW))$ following [25]. Both truncations and ReLUs are element-wise and hence, their communication cost scales linearly with the input feature size, i.e., $O(HWC)$. Therefore, for networks with a large number of channels, the pre-processing communication becomes even more dominating.

Since the communication is dominated by the multiplications in convolution layers, *our intuition is to reduce the multiplications by conducting more local additions for better communication efficiency*. Meanwhile, as in Figure 2, MobileNetV2 are much more communication efficient compared to ResNets with a similar accuracy, and hence, becomes the focus of our paper.

**Observation 3: ReLU pruning and network re-parameterization can simultaneously reduce the online and total communication of all operations.** Existing network optimizations either linearize the ReLUs selectively [26, 7, 40, 37, 6, 32] or prune the whole channel or layer [36]. Selective ReLU linearization usually achieves a better accuracy but achieves limited communication reduction. In contrast, channel-wise or layer-wise pruning can simultaneously reduce the communication for all the operators but usually suffers from a larger accuracy degradation [26]. We observe for an inverted residual block in MobileNetV2, as shown in Figure 3, if both of the two ReLUs are removed, the block can be merged into a dense convolution after training. *During training, this preserves the benefits of over-parameterization shown in RepVGG [14] to protect network accuracy, while during*

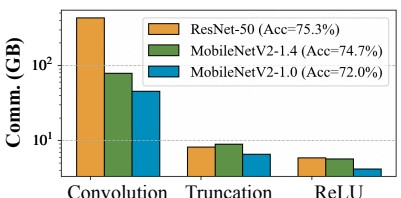

Figure 2: Comparison of communication breakdown between ResNet-50 and MobileNetV2 on ImageNet.

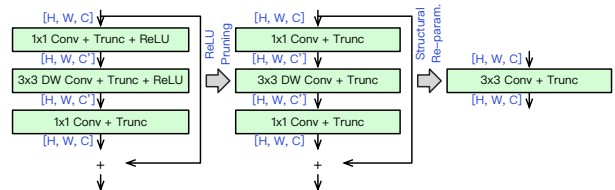

Figure 3: Illustration of ReLU pruning and network re-parameterization for communication reduction. DW means depth-wise and $1 \times 1$ Conv is also called point-wise (PW) convolution.

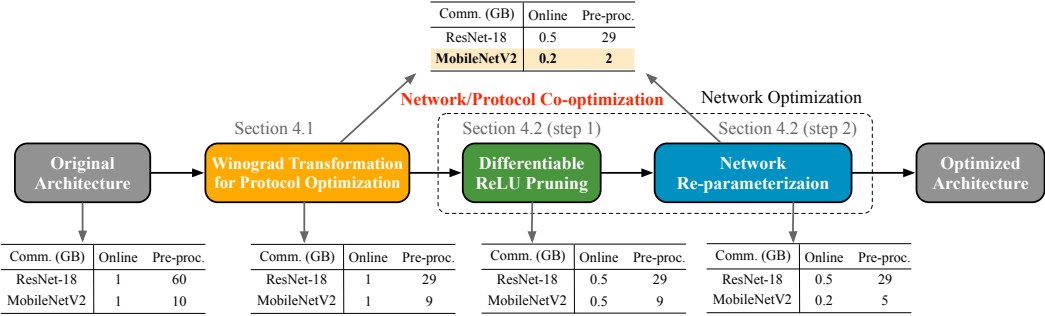

Figure 4: Overview of CoPriv and the communication cost after each optimization step. The example of the communication cost is measured on the CIFAR-100 dataset.

*inference, the dense convolution is more compatible with our optimized protocol and can reduce communication of truncations as well.*

## 4   CoPriv: A New Paradigm Towards Efficient Private Inference

In this section, we present CoPriv, a protocol-network co-optimization framework for communication-efficient 2PC-based inference. The overview of CoPriv is shown in Figure 4. Given the original network, we first leverage the Winograd transformation with DNN-aware optimization to reduce the communication when computing 3-by-3 convolutions without any accuracy impact (Section 4.1). The optimized transformation enables to reduce the total communication of ResNet-18 by $2.1 \times$. However, the communication reduction of MobileNetV2 is limited as the Winograd transformation can only be applied to the 3-by-3 depth-wise convolution, which only contributes to a small portion of the total communication. Hence, for MobileNetV2, we propose an automatic and differentiable ReLU pruning algorithm (Section 4.2). By carefully designing the pruning pattern, CoPriv enables further network re-parameterization of the pruned networks, which reduces the online and total communication by $5 \times$ and $5 \times$ in the example, respectively, combining with our optimized convolution protocol.

### 4.1   Winograd Transformation for Protocol Optimization

We propose to leverage the Winograd transformation to reduce the communication when computing a convolution. Following Eq. 1, for each 2D output tile, because $A, B, G$ are known publicly, the filter transformation $GWG^\top$ and feature transformation $B^\top X B$ can be computed locally on the server and client [17]. Then, EWMM is performed together by the server and the client through communication while the final output transformation $A^\top[\cdot]A$ can be computed locally as well.

For each output tile, as $(m + r - 1)^2$ multiplications are needed and there is no shared multiplier for batch optimization described in Section 2.2, the communication cost is $O(\lambda(m+r-1)^2)$ [2]. Consider there are in total $T = \lceil H/m \rceil \times \lceil W/m \rceil$ tiles per output channel, $C$ input channels, and $K$ output channels, the communication complexity of a convolution layer now becomes $O(\lambda TCK(m+r-1)^2)$. To reduce the enormous communication, we present the following optimizations.

---

[2]For simplicity, we omit the bit precision $l$ for all the downstream analysis.

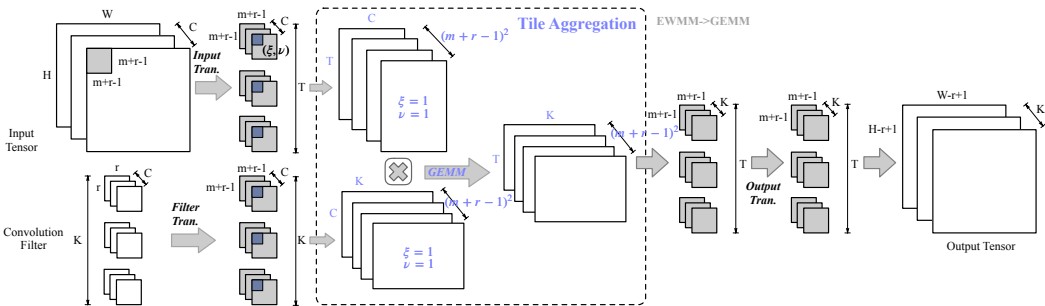

Figure 5: Winograd transformation procedure with tile aggregation.

Table 3: Communication complexity of different convolution types and the measured communication for a ResNet-18 block with $14\times14\times256$ feature dimension.

| Conv. Type | Comm. Complexity | ResNet Block Comm. (GB) |
|---|---|---|
| Regular Conv. | $O(CKr^2(\lambda + HW))$ | 23.74 (1×) |
| EWMM-based Winograd Conv. | $O(\lambda CKT(m+r-1)^2)$ | 256 (10.78× ↑) |
| + Tile Aggregation (GEMM-based) | $(m+r-1)^2CT(\lambda+K)$ | 10.45 (2.27× ↓) |
| + DNN-aware Sender Selection | $\min[(m+r-1)^2CT(\lambda+K),$ $(m+r-1)^2CK(\lambda+T)]$ | 7.76 (3.06× ↓) |

**Communication reduction with tile aggregation**   While the Winograd transformation helps to reduce the total number of multiplications, in Table 3, we observe the communication for a ResNet block actually increases. This is because the baseline protocol can leverage the batch optimization mentioned in Section 2.2 to reduce the communication complexity. To further optimize the Winograd-based convolution protocol to adapt to private inference, we observe the following opportunities for the batch optimization: first, each transformed input tile needs to multiply all the $K$ filters; secondly, each transformed filter needs to multiply all the transformed input tiles for a given channel. Hence, for the $t$-th tile in $f$-th output channel, we re-write Eq. 1 as follows:

$$Y_{f,t} = A^\top[\sum_{c=1}^{C} U_{f,c} \odot V_{c,t}]A \quad 1 \le f \le K, 1 \le t \le T,$$

where $U_{f,c}$ denotes the transformed filter correspnding to the $f$-th output channel and $c$-th input channel and $V_{c,t}$ denotes the transformed input of the $t$-th tile in the $c$-th input channel. Then, consider each pixel location, denoted as $(\xi, \nu)$, within the tile separately, yielding:

$$Y_{f,t}^{(\xi,\nu)} = A^\top[U_{f,c}^{(\xi,\nu)}V_{c,t}^{(\xi,\nu)}]A \quad 1 \le f \le K, 1 \le t \le T, 1 \le \xi, \nu \le m+r-1$$

We illustrate this transformation procedure in Figure 5. Re-writing the EWMM into general matrix multiplication (GEMM) enables us to fully leverage the batch optimization and reduce the communication complexity as shown in Table 3. We can now reduce the communication of a ResNet block by $2.27\times$.

**DNN-aware adaptive convolution protocol**   When executing the convolution protocol, both the server and the client can initiate the protocol. Because the input feature map and the filter are of different dimensions, we observe the selection of protocol intializer impacts the communication round as well as the communication complexity. While CrypTFlow2 always selects the server to initialize the protocol, we propose DNN architecture-aware convolution protocol to choose between the server and the client adaptively based on the layer dimensions to minimize the communication complexity. As shown in Table 3, the communication of the example ResNet block can be further reduced by $1.35\times$ with the adaptive protocol.

## 4.2   Differentiable ReLU Pruning and Network Re-Parameterization

We now describe our network optimization algorithm to further reduce both pre-processing and online communication for MobileNetV2. *The core idea is to simultaneously remove the two ReLUs*

*within an inverted residual block together, after which the entire block can be merged into a dense convolution layer to be further optimized with our Winograd-based protocol* as in Figure 3. For the example in Figure 4, both the pre-processing and online communication for MobileNetV2 can be reduced by $5\times$. To achieve the above goal, the remaining questions to answer include: 1) which ReLUs to remove and how to minimize the accuracy degradation for the pruned networks, and 2) how to re-parameterize the inverted residual block to guarantee functional correctness. Hence, we propose the following two-step algorithm.

**Step 1: communication-aware differentiable ReLU pruning**  To identify "unimportant" activation functions, CoPriv leverages a differentiable pruning algorithm. Specifically, CoPriv assigns an architecture parameter $\alpha(0 \leq \alpha \leq 1)$ to measure the importance of each ReLU. During pruning, the forward function of a ReLU now becomes $\alpha \cdot \text{ReLU}(x) + (1 - \alpha) \cdot x$. CoPriv jointly learns the model weights $\theta$ and the architecture parameters $\alpha$. Specifically, given a sparsity constraint $s$, we propose the one-level optimization formulated as follows:

$$\min_{\theta,\alpha} \mathcal{L}_{CE} + \mathcal{L}_{comm} \quad \text{s.t.} \quad ||\alpha||_0 \leq s, \tag{2}$$

where $\mathcal{L}_{CE}$ is the task-specific cross entropy loss and $\mathcal{L}_{comm}$ is the communication-aware regularization to focus the pruning on communication-heavy blocks and is defined as

$$\mathcal{L}_{comm} = \sum_i \alpha_i(\text{Comm}_i(\alpha_i = 1) - \text{Comm}_i(\alpha_i = 0)),$$

where $\text{Comm}_i(\alpha_i = 1)$ is the communication to compute the $i$-th layer when $\alpha_i = 1$. We also show the effectiveness of communication-aware regularization in the experimental results. To ensure the network after ReLU pruning can be merged, two ReLUs within the same block share the same $\alpha$. During training, in the forward process, only the top-$s$ $\alpha$'s are activated (i.e., $\alpha = 1$) while the remainings are set to 0. In the backpropagation process, $\alpha$'s are updated via straight-through estimation (STE) [4].

Once the pruning finishes, the least important ReLUs are removed and we perform further finetuning to improve the network accuracy. Specifically, in CoPriv, we leverage knowledge distillation (KD) [21] to guide the finetuning of the pruned network.

**Step 2: network re-parameterization**  The removal of ReLUs makes the inverted residual block linear, and thus can be further merged together into a single dense convolution. Motivated by the previous work of structural re-parameterization [16, 14, 13, 11, 10, 12], we describe the detailed re-parameterization algorithm in Appendix C. The re-parameterized convolution has the same number of input and output channels as the first and last point-wise convolution, respectively. Its stride equals to the stride of the depth-wise convolution.

## 5 Experiments

### 5.1 Experimental Setup

We adopt CypTFlow2 [43] protocol for the 2PC-based inference, and we measure the communication and latency under a LAN setting [43] with 377 MBps bandwidth and 0.3ms echo latency. For Winograd, we implement $F(2 \times 2, 3 \times 3)$ and $F(4 \times 4, 3 \times 3)$ transformation for convolution with stride of 1 and $F(2 \times 2, 3 \times 3)$ transformation when stride is 2 [23]. We apply CoPriv to MobileNetV2 with different width multipliers on CIFAR-100 [30] and ImageNet [9] datasets. The details of our experimental setups including the private inference framework, the implementation of the Winograd-based convolution protocol and the training details are available in Appendix B.

### 5.2 Micro-Benchmark on the Convolution Protocol with Winograd Transformation

We first benchmark the communication reduction of the proposed convolution protocol based on Winograd transformation. The results on ResNet-18 and ResNet-32 are shown in Figure 6(a) and (b). As can be observed, the proposed protocol consistently reduces the communication to compute the convolutions. While the degree of communication reduction depends on the layer dimensions, on average, $2.1\times$ reduction is achieved for both ResNet-18 and ResNet-32. We also

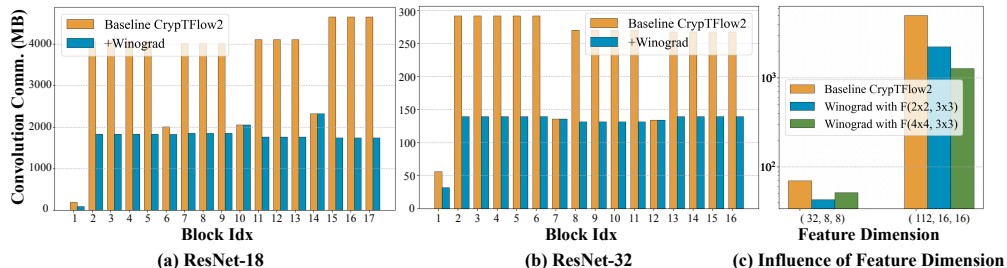

Figure 6: Communication of the convolution protocol with Winograd transformation on (a) ResNet-18 and (b) ResNet-32 on CIFAR-100; (c) comparison between $F(2 \times 2, 3 \times 3)$ and $F(4 \times 4, 3 \times 3)$ with different feature dimensions.

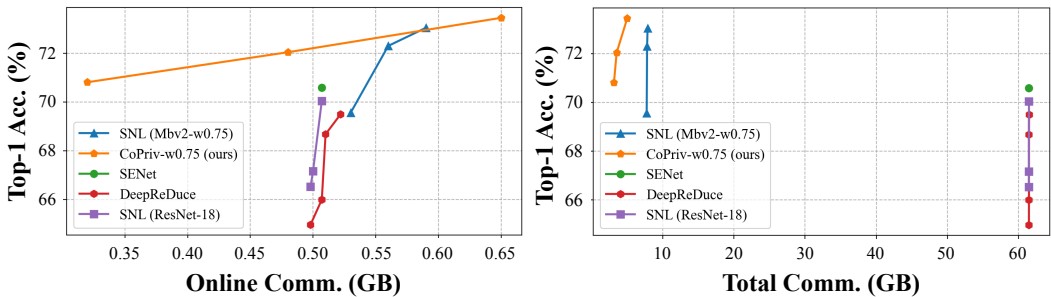

Figure 7: Comparison with efficient ReLU-optimized methods on CIFAR-100 dataset.

benchmark the Winograd-based protocol with output tile sizes of 2 and 4, i.e., $F(2 \times 2, 3 \times 3)$ and $F(4 \times 4, 3 \times 3)$. Figure 6(c) shows that when the feature resolution is small, a small tile size achieves more communication reduction while larger resolution prefers a larger tile size. Hence, for the downstream experiments, we use $F(2 \times 2, 3 \times 3)$ for CIFAR-100 and $F(4 \times 4, 3 \times 3)$ for ImageNet [9] dataset, respectively.

## 5.3 Benchmark with ReLU-Optimized Networks on CIFAR-100

We benchmark CoPriv with different network optimization algorithms that focus on reducing ReLU counts, including DeepReDuce [26], SENet [32] and SNL [7], on the CIFAR-100 dataset. As DeepReDuce, SENet and SNL all use ResNet-18 as the baseline model, we also train SNL algorithm on the MobileNetV2-w0.75 for a comprehensive comparison.

**Results and analysis** From Figure 7, we make the following observations: 1) previous methods, including DeepReDuce, SENet, SNL, do not reduce the pre-processing communication and their online communication reduction quickly saturates with a notable accuracy degradation; 2) CoPriv achieves SOTA Pareto front of accuracy and online/total communication. Specifically, CoPriv outperforms SNL on ResNet-18 with 4.3% higher accuracy as well as 1.55× and 19.5× online and total communication reduction, respectively. When compared with SNL on MobileNetV2-w0.75, CoPriv achieves 1.67% higher accuracy with 1.66× and 2.44× online and total communication reduction, respectively.

## 5.4 Benchmark on ImageNet

We also benchmark CoPriv on the ImageNet dataset with the following baselines: lightweight mobile networks like MobileNetV3 [22] with different capacities, ReLU-optimized networks, including SNL [7] and SENet [32], and SOTA pruning methods, including uniform pruning and MetaPruning [36]. We train both our CoPriv and SNL on the MobileNetV2-w1.4 with self distillation.

**Results and analysis** From the results shown in Figure 8, we observe that 1) compared with SNL on MobileNetV2-w1.4, CoPriv achieves 1.4% higher accuracy with 9.98× and 3.88× online and total communication reduction, respectively; 2) CoPriv outperforms 1.8% higher accuracy compared

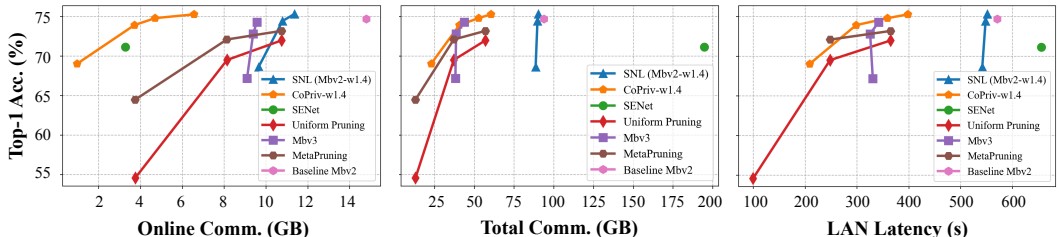

Figure 8: Comparison with SOTA efficient ConvNets, ReLU-optimized methods, channel-wise pruning MetaPruning [36] and uniform pruning on ImageNet dataset. We mark the four points of CoPriv as CoPriv-w1.4-A/B/C/D from left to right, respectively.

Table 4: Ablation study of our proposed optimizations in CoPriv on CIFAR-100. Two sparsity constraints are on the two sides of "/", respectively.

| Model | Online Communication (GB) | Total Communication (GB) |
|---|---|---|
| Baseline SNL (ResNet-18) | 0.507 | 61.45 |
| +Winograd | 0.507 | 33.84 |
| Baseline SNL (MobileNetV2) | 0.530 | 7.710 |
| +Winograd | 0.530 | 7.132 |
| MobileNetV2+Pruning | 0.606 / 0.582 | 7.806 / 7.761 |
| +Re-parameterization | 0.331 / 0.260 | 4.197 / 3.546 |
| +Winograd (CoPriv) | 0.331 / 0.260 | 3.154 / 2.213 |

with MobileNetV3-Small-1.0 while achieving $9.41\times$ and $1.67\times$ online and total communication reduction; 3) CoPriv demonstrates its strong scalability for communication optimization. Compared to the baseline MobileNetV2-w1.4, CoPriv achieves $2.52\times$ and $1.9\times$ online and total communication reduction, respectively without compromising the accuracy; 4) when compared with SOTA pruning methods, for a high communication budget, CoPriv-w1.4-B achieves 1.6% higher accuracy with $2.28\times$ and $1.08\times$ online and total communication reduction, compared with MetaPruning-1.0$\times$; for a low communication budget, compared with MetaPruning-0.35$\times$, CoPriv-w1.4-D achieves 5.5% higher accuracy with $3.87\times$ online communication reduction.

## 5.5 Ablation Study

**Effectiveness of different optimizations in CoPriv** To understand how different optimizations help improve communication efficiency, we add our proposed protocol optimizations step by step on both SNL and CoPriv, and present the results in Table 4. According to the results, we find that 1) our Winograd-based convolution protocol consistently reduces the total communication for both networks; 2) when directly applying the Winograd transformation on MobileNetV2, less than 10% communication reduction is achieved as Winograd transformation only helps the depth-wise convolution, which accounts for only a small portion of pre-processing communication; 3) compared with SNL, CoPriv achieves higher accuracy with much lower communication. The findings indicate that all of our optimizations are indispensable for improving the communication efficiency.

**Block-wise communication comparison and visualization** We compare and visualize the block-wise communication reduction of CoPriv on MobileNetV2-w0.75 on the CIFAR-100 dataset. From Figure 9, it is clear that our adaptive convolution protocol is effective for communication optimization, and different layers benefit from CoPriv differently. More specifically, for the total communication, block # 4 benefits more from the Winograd transformation over the network re-parameterization, while block # 16 benefits from both the re-parameterization and the adaptive convolution protocol. Both pruning and re-parameterization are important for online communication as they remove the communication of ReLU and truncation, respectively. The results demonstrate the importance of all the proposed optimization techniques.

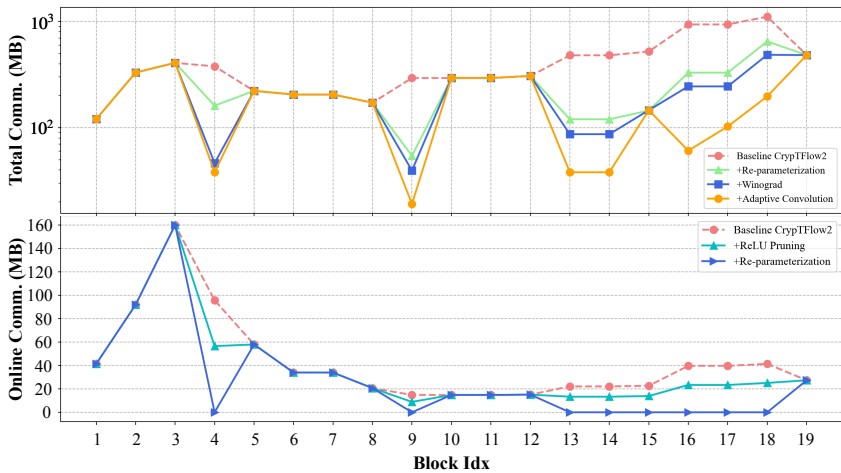

Figure 9: Block-wise visualization of online and total communication with different techniques on CoPriv-w0.75 on CIFAR-100.

## 5.6 Effectiveness of Communication-Aware Regularization

To show the effectiveness and importance of introducing communication-aware regularization $\mathcal{L}_{comm}$ formulated in Eq. 2, we compare our pruning method with $\mathcal{L}_{comm}$ and pruning methods of SNL [7] and SENet [32] without $\mathcal{L}_{comm}$ in Figure 10. As we can observe, $\mathcal{L}_{comm}$ indeed helps to focus the pruning on the later layers, which incur more communication cost, and penalizes the costly blocks (e.g., block # 16 and # 18). In contrast, SNL does not take communication cost into consideration and SENet (ReLU sensitivity based) focuses on pruning early layers with more ReLUs, both of which incur huge communication overhead.

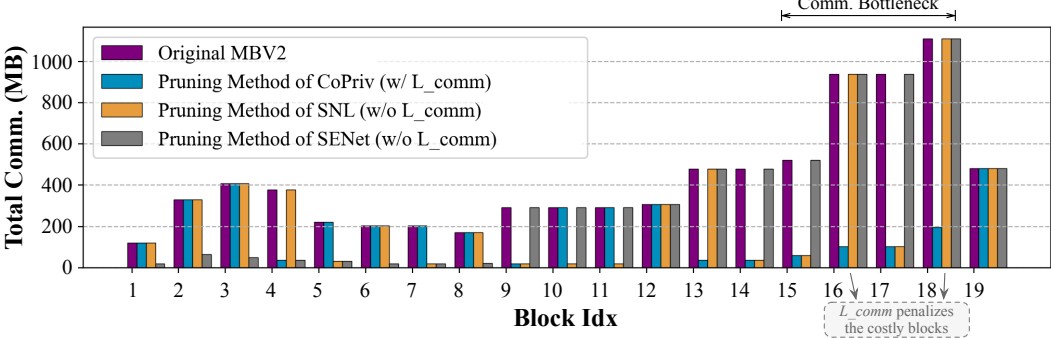

Figure 10: Comparison of different pruning method and the influence of $\mathcal{L}_{comm}$ during the search in each block (ReLU budget is set to 60% in this example).

## 6 Conclusion

In this work, we propose a network/protocol co-optimization framework, CoPriv, that simultaneously optimizes the pre-processing and online communication for the 2PC-based private inference. CoPriv features an optimized convolution protocol based on Winograd transformation and leverages a series of DNN-aware protocol optimization to improve the efficiency of the pre-processing stage. We also propose a differentiable communication-aware ReLU pruning algorithm with network re-parameterization to further optimize both the online and pre-processing communication induced by ReLU, truncation and convolution. With extensive experiments, CoPriv consistently reduces both online and total communication without compromising the accuracy compared with different prior-art efficient networks and network optimization algorithms.

## Acknowledgement

This work was supported in part by the NSFC (62125401) and the 111 Project (B18001).

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

# A    Related Works

Private inference has been a promising solution to protect both data and model privacy during deep learning inference. In recent years, there has been an increasing amount of literature on efficient private inference. According to the optimization technique, these works can be categorized into three types, i.e., 1) protocol optimization; 2) network optimization; and 3) joint optimization.

In protocol optimization, ABY [8] provides a highly efficient conversion between arithmetic sharing, boolean sharing and Yao's sharing, and construct mixed protocols. As an extension, ABY3 [38] switches back and forth between three secret sharing schemes using three-party computation (3PC). CypTFlow2 [43] proposes a new protocol for secure and comparison and division which enables effecient non-linear operations such as ReLU. SiRNN [42] further proposes 2PC protocols for bitwidth extension, mixed-precision linear and non-linear operations. CrypTen [29] proposes a software framework that provides a flexible machine learning focused API. More recently, SecFloat [41] proposes the crypto-friendly precise functionalities to build a library for 32-bit single-precision floating-point operations and math functions. These works lack consideration for neural network architecture and has limited communication reduction.

In network optimization, DeepReDuce [26] proposes to manually remove ReLUs with a three-step optimization pipline. SNL [7] proposes ReLU-aware optimization that leverages gradient-based NAS to selectively linearize a subset of ReLUs. CryptoNAS [19] uses ReLU budget as a proxy and leverages NAS to tailor ReLUs. PolyMPCNet [40] and SAFENet [37] replace ReLUs with MPC-friendly polynomial, while Sphynx [6] proposes an MPC-friendly ReLU-efficient micro-search space. SENet [32] innovatively measures the ReLU importance via layer pruning sensitivity and automatically optimize the network to meet the target ReLU budget. DeepReShape [26] finds that wider networks are more ReLU-efficient than the deeper ones and designs ReLU-efficient baseline networks with with FLOPs-ReLU-Accuracy balance. For Transformer-based models, MPCFormer [35] proposes to simplify Softmax by replacing exponential with a more MPC-friendly quadratic operation. MPCViT [51] proposes to mix both high accuracy and high efficiency attention variants to accelerate private Vision Transformer (ViT) inference. Network optimization mainly focuses on ReLU reduction which dominates the online communication, but total communication including convolution and truncation cannot be optimized.

Unluckily, only using either protocol or network optimization just leads to limited efficiency improvement. Delphi [48] jointly optimizes cryptographic protocols and network by gradually replacing ReLU with quadratic approximation. COINN [25] simultaneously optimizes quantized network and protocols with ciphertext-aware quantization and automated bitwidth configuration. Recently, [17] proposes to use Winograd convolution for reducing the number of multiplications and design the efficient convolution operation to reduce the communication cost. However, it does not take private inference into consideration for Winograd algorithm, and still suffers tremendous communication overhead. In this work, we jointly optimize the network and protocol and fully consider their coupling properties.

# B    Details of Experimental Setup

**Private inference framework**    CoPriv is built based on CypTFlow2 [43] protocol for private inference. We leverage the Athos [43] tool chain to convert both input and weight into fixed-point with the bit-width 41 and scale 12. We measure the communication and latency under a LAN setting [43] with 377 MBps bandwidth and 0.3ms echo latency. All of our experiments are evaluated on the Intel Xeon Gold 5220R CPU @ 2.20GHz.

**Implementation of Winograd-based convolution protocol**    The convolution protocol with Winograd transformation and optimization is implemented in C++ with Eigen and Armadillo matrix calculation library [45] in the CrypTFlow2 [43] framework. We implement $F(2 \times 2, 3 \times 3)$ and $F(4 \times 4, 3 \times 3)$ transformation for convolution with stride of 1 and $F(2 \times 2, 3 \times 3)$ transformation when stride is 2 [23]. For CIFAR-100 dataset, we use $F(2 \times 2, 3 \times 3)$ transformation as the image resolution is small and for ImageNet dataset, we use $F(4 \times 4, 3 \times 3)$. We only apply $F(2 \times 2, 3 \times 3)$ for stride of 2 on ImageNet dataset. When evaluating CoPriv, we determine the optimal sender according to the analysis in Table 3 before inference. Winograd implementation enables us to measure the communication cost and latency of each convolution module.

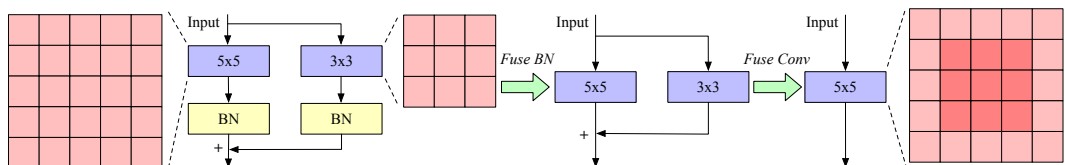

Figure 11: An example of re-parameterization for batch normalization and convolution filter.

**Networks and datasets** We apply our proposed CoPriv to the widely used lightweight mobile network MobileNetV2 [46] with different width multipliers, e.g., 0.75, 1.0 and 1.4 to trade off the model accuracy and efficiency. We evaluate the top-1 accuracy and online and total communication on both CIFAR-100 and ImageNet [9] dataset.

**Differentiable pruning and finetuning setups** We first search and prune redundant ReLUs for 10 epochs and then finetune the pruned network for 180 epochs with stochastic gradient descent (SGD) optimizer [2], cosine learning scheduler and initial learning rate of 0.1. During finetuning stage, we train our proposed CoPriv with knowledge distillation to boost its performance.

## C   Network Re-Parameterization Algorithm

Network (or structural) re-parameterization is a useful technique proposed by RepVGG [14], and is extended to [11, 10, 13, 16, 12]. The core idea of re-parameterization is to decouple the training-time architecture (with high performance and low efficiency) and inference-time network architecture (with equivalent performance and high efficiency). Re-parameterization is realized by converting one architecture to another via equivalently merging parameters together. Therefore, during inference time, the network architecture is not only efficient but also has the same high performance as the training-time architecture. Figure 11 is an simple example of re-parameterization.

In this work, we can also leverage this technique to merge adjacent convolutions together after ReLU removal. For the network re-parameterization mentioned in Section 4.2, here we provide the following detailed algorithm 1 to equivalently merge the inverted residual block into a single dense convolution as shown in Figure 3. With the help of network re-parameterization, we further optimize the total communication including convolution and truncation.

---

**Algorithm 1:** Network Re-parameterization for Inverted Residual Block

**Input** : An inverted residual block with weights $W_{1\times1}$, $W_{3\times3}$, and $W'_{1\times1}$. The number of input and output channels $N_{in}, N_{out}$. The size of re-parameterized weights $r$.

**Output** : Regular convolution with re-parameterized weights $W_r$.

1  $W_r = $ torch.eye($N_{in}$);
2  $W_r = W_r$.unsqueeze($2$).unsqueeze($2$);
3  $W_r = $ torch.nn.functional.pad($W_r$, *pad*=($\frac{r-1}{2}$, $\frac{r-1}{2}$, $\frac{r-1}{2}$, $\frac{r-1}{2}$));
4  $W_r = $ torch.nn.functional.conv2d($W_r$, $W_{1\times1}$);
5  $W_r = $ torch.nn.functional.conv2d($W_r$, $W_{3\times3}$, *padding*=$\frac{r-1}{2}$);
6  $W_r = $ torch.nn.functional.conv2d($W_r$, $W'_{1\times1}$);
7  $W_{res} = $ torch.zeros($N_{out}, N_{in}, r, r$);
8  **for** $i \in [0, \ldots, N_{out} - 1]$ **do**
9  $\quad \lfloor \; W_{res}[i, i, \lfloor r/2 \rfloor, \lfloor r/2 \rfloor] = 1$;
10  $W_r = W_r + W_{res}$;
11  **return** $W_r$;

---

## D   Details of Winograd Convolution

### D.1   Comparison between Regular Convolution and Winograd Convolution

To help readers better understand the multiplication reduction of Winograd convolution, we demonstrate regular convolution and Winograd convolution in Figure 12. Given an input $I \in \mathbb{R}^{4\times4}$ and

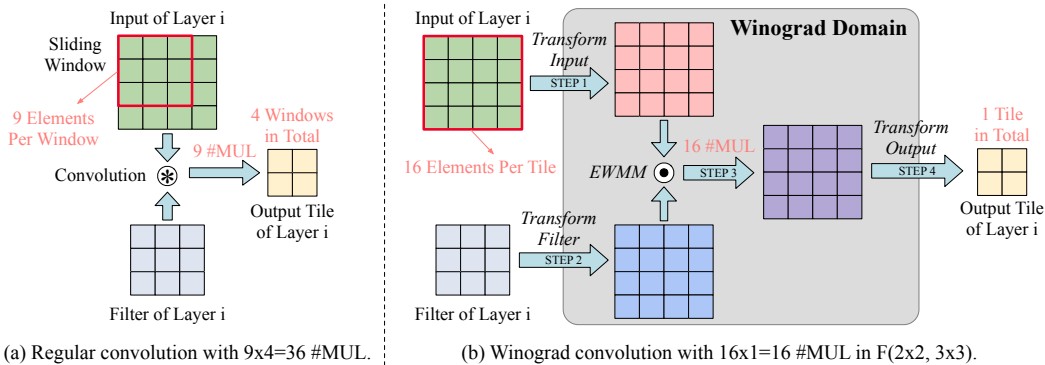

|  |  |
|---|---|
| (a) Regular convolution with 9x4=36 #MUL. | (b) Winograd convolution with 16x1=16 #MUL in F(2x2, 3x3). |

Figure 12: Comparison between (a) regular convolution and (b) Winograd convolution.

a filter $F \in \mathbb{R}^{3 \times 3}$, regular convolution requires $9 \times 4 = 36$ times multiplications (implemented using GEMM with im2col algorithm [5]) while $F(2 \times 2, 3 \times 3)$ Winograd transformation only requires $16 \times 1 = 16$ times multiplications (EWMM), which achieves $2.25\times$ reduction. Moreover, $F(4 \times 4, 3 \times 3)$ with a larger tile size, i.e., 6 can further achieve $4\times$ multiplication reduction. The improvement gets benefit from the Winograd's ability to convert im2col to EWMM and calculate the whole tile in Winograd domain at once.

## D.2 Details of Input Tiling and Padding

Given a large 2D input $I \in \mathbb{R}^{l \times l}$, where $l > m + r - 1$, the core technique for ensuring the equivalence of regular convolution and Winograd convolution is input tiling and padding. The output size $l' = l - r + 1$, the input tile size $n = m + r - 1$ and the total tile number $T$ per channel is computed as

$$T = \lceil \frac{l'}{n} \rceil^2 = \lceil \frac{l - r + 1}{m + r - 1} \rceil^2,$$

where $\lceil \cdot \rceil$ denotes taking the upper bound value. For each tile, Winograd convolution is individually performed and results an output tile with $m \times m$ size. After all the tiles are computed with Winograd convolution, the output tiles are concatenated together to form the final output.

For some input size, the input cannot be covered by tiles. For instance, when leveraging $F(2 \times 2, 3 \times 3)$ on the input $I \in \mathbb{R}^{7 \times 7}$, the rightmost and bottom pixels cannot be divided into a complete tile. To solve this problem, we pad these positions with 0 to enable the tiles totally cover the whole input. The correctness and equivalence can be proved with Eq. 1. Also, [17] shows the overhead caused by padding is negligible.

## D.3 Support for Stride of 2 Winograd Convolution

Conventional Winograd convolution only supports stride $s = 1$ convolution filter. However, in recent efficient neural networks, e.g., MobileNetV2, EfficientNet has several stride of 2 layers to reduce the feature map size by half. To enable extreme optimization for efficient networks, we introduce $F(2 \times 2, 3 \times 3)$ for stride of 2 Winograd convolution for private inference.

There are various methods to construct stride of 2 Winograd kernel such as dividing input and convolution filter into different groups [50]. However, it is not a simple way to implement stride of 2 Winograd kernel. [23] is an extremely convenient method using unified transformation matrices.

Based on [23], even positions of input and filter are computed by $F(2, 2)$ while odd positions are computed by regular convolution. Transformation matrices are derived as follows and can be computed using Eq. 1:

$$B^{\top} = \begin{bmatrix} 1 & 0 & -1 & 0 & 0 \\ 0 & 1 & 0 & 0 & 0 \\ 0 & 0 & 1 & 0 & 0 \\ 0 & 0 & 0 & 1 & 0 \\ 0 & 0 & -1 & 0 & 1 \end{bmatrix}, \quad G = \begin{bmatrix} 1 & 0 & 0 \\ 0 & 1 & 0 \\ 1 & 0 & 1 \\ 0 & 1 & 0 \\ 0 & 0 & 1 \end{bmatrix}, \quad A^{\top} = \begin{bmatrix} 1 & 1 & 1 & 0 & 0 \\ 0 & 0 & 1 & 1 & 1 \end{bmatrix}.$$

**Correctness analysis.** Here, we take a 1D algorithm as an example to prove the correctness Winograd convolution for stride of 2. The algorithm can be nested with itself to obtain a 2D algorithm [34].

Given input $X$ and filter $F$ as

$$X = \begin{bmatrix} x_0 \\ x_1 \\ x_2 \\ x_3 \\ x_4 \end{bmatrix}, \quad F = \begin{bmatrix} y_0 \\ y_1 \\ y_2 \end{bmatrix}, \quad Y = X \circledast F = \begin{bmatrix} z_0 \\ z_1 \end{bmatrix}.$$

First, we calculate regular convolution with stride of 2 using im2col algorithm [5] as

$$Y_1 = \begin{bmatrix} x_0 & x_1 & x_2 \\ x_2 & x_3 & x_4 \end{bmatrix} \cdot \begin{bmatrix} y_0 \\ y_1 \\ y_2 \end{bmatrix} = \begin{bmatrix} x_0 y_0 + x_1 y_1 + x_2 y_2 \\ x_2 y_0 + x_3 y_1 + x_4 y_2 \end{bmatrix},$$

thus, $z_0 = x_0 y_0 + x_1 y_1 + x_2 y_2$ and $z_1 = x_2 y_0 + x_3 y_1 + x_4 y_2$.

Then, we calculate Winograd convolution for stride of 2 as

$$Y = A^\top \cdot [(GF) \odot (B^\top X)],$$

and then

$$Y_2 = \begin{bmatrix} 1 & 1 & 1 & 0 & 0 \\ 0 & 0 & 1 & 1 & 1 \end{bmatrix} \cdot [(\begin{bmatrix} 1 & 0 & 0 \\ 0 & 1 & 0 \\ 1 & 0 & 1 \\ 0 & 1 & 0 \\ 0 & 0 & 1 \end{bmatrix} \cdot \begin{bmatrix} y_0 \\ y_1 \\ y_2 \end{bmatrix}) \odot (\begin{bmatrix} 1 & 0 & -1 & 0 & 0 \\ 0 & 1 & 0 & 0 & 0 \\ 0 & 0 & 1 & 0 & 0 \\ 0 & 0 & 0 & 1 & 0 \\ 0 & 0 & -1 & 0 & 1 \end{bmatrix} \cdot \begin{bmatrix} x_0 \\ x_1 \\ x_2 \\ x_3 \\ x_4 \end{bmatrix})],$$

and further simplify the calculation as

$$Y_2 = \begin{bmatrix} 1 & 1 & 1 & 0 & 0 \\ 0 & 0 & 1 & 1 & 1 \end{bmatrix} \cdot [(\begin{bmatrix} y_0 \\ y_1 \\ y_0 + y_2 \\ y_1 \\ y_2 \end{bmatrix}) \odot (\begin{bmatrix} x_0 - x_2 \\ x_1 \\ x_2 \\ x_3 \\ x_4 - x_2 \end{bmatrix})] = \begin{bmatrix} 1 & 1 & 1 & 0 & 0 \\ 0 & 0 & 1 & 1 & 1 \end{bmatrix} \cdot \begin{bmatrix} x_0 y_0 - x_2 y_0 \\ x_1 y_1 \\ x_2 y_0 + x_2 y_2 \\ x_3 y_1 \\ x_4 y_2 - x_2 y_2 \end{bmatrix},$$

therefore, the convolution result is

$$Y_2 = \begin{bmatrix} x_0 y_0 + x_1 y_1 + x_2 y_2 \\ x_2 y_0 + x_3 y_1 + x_4 y_2 \end{bmatrix} = Y_1.$$

### D.4 Transformation Matrices for Winograd Convolution

We provide the transformation matrices $A, B, G$ for $F(2 \times 2, 3 \times 3)$ and $F(4 \times 4, 3 \times 3)$ Winograd transformation based on polynomial Chinese remainder theorem (CRT) or Lagrange interpolation [34].

For $F(2 \times 2, 3 \times 3)$, we have

$$B^\top = \begin{bmatrix} 1 & 0 & -1 & 0 \\ 0 & 1 & 1 & 0 \\ 0 & -1 & 1 & 0 \\ 0 & 1 & 0 & -1 \end{bmatrix}, \quad G = \begin{bmatrix} 1 & 0 & 0 \\ 1/2 & 1/2 & 1/2 \\ 1/2 & -1/2 & 1/2 \\ 0 & 0 & 1 \end{bmatrix}, \quad A^\top = \begin{bmatrix} 1 & 1 & 1 & 0 \\ 0 & 1 & -1 & -1 \end{bmatrix}.$$

For $F(4 \times 4, 3 \times 3)$, we have

$$B^\top = \begin{bmatrix} 4 & 0 & -5 & 0 & 1 & 0 \\ 0 & -4 & -4 & 1 & 1 & 0 \\ 0 & 4 & -4 & -1 & 1 & 0 \\ 0 & -2 & -1 & 2 & 1 & 0 \\ 0 & 2 & -1 & -2 & 1 & 0 \\ 0 & 4 & 0 & -5 & 0 & 1 \end{bmatrix}, \quad G = \begin{bmatrix} 1/4 & 0 & 0 \\ -1/6 & -1/6 & -1/6 \\ -1/6 & 1/6 & -1/6 \\ 1/24 & 1/12 & 1/6 \\ 1/24 & -1/12 & 1/6 \\ 0 & 0 & 1 \end{bmatrix},$$

$$A^\top = \begin{bmatrix} 1 & 1 & 1 & 1 & 1 & 0 \\ 0 & 1 & -1 & 2 & -2 & 0 \\ 0 & 1 & 1 & 4 & 4 & 0 \\ 0 & 1 & -1 & 8 & -8 & 1 \end{bmatrix}.$$

The correctness analysis is the same with Section D.3.

# E  Comparison with More Related Work

[33] is a recent work that aims at optimizing the network architecture for latency-efficient private inference. Both CoPriv and [33] aim to reduce the number of ReLUs and the depth of the whole network in order to improve the efficiency. However, they have totally different motivations and methods. 1) Motivations: [33] still regards ReLU as the main latency bottleneck and removes convolution layers to reduce computation. In contrast, CoPriv fuses neighboring convolution layers to better leverage our Winograd-based optimization for communication reduction of all operators. The difference in motivation leads to different criterion when selecting convolutions to remove. [33] selects convolutions based on ReLU sensitivity [32] while CoPriv simultaneously considers both accuracy and communication cost. 2) Methods: [33] first determines which convolutions to remove based on ReLU sensitivity [32] and then, uses the gated branching method for training. In contrast, CoPriv simultaneously train the architecture parameters with the model weights, and the equivalent re-parameterization is conducted post training, enabling us to leverage the benefits of over-parameterization during training shown in RepVGG [14]. Quantitatively, we compare CoPriv with [33] in Table 5. CoPriv achieves much lower online and total communication while achieving higher accuracy compared to [33].

Table 5: Accuracy and communication comparison between CoPriv and [33].

| Method | Top-1 Acc. (%) | Online Comm. (GB) | Total Comm. (GB) |
|---|---|---|---|
| [33] | 69.10 | 0.81 | 46.5 |
| CoPriv (ours) | 70.58 | 0.43 | 5.14 |

# F  More Detailed Comparison with Prior-Art Methods

As a supplement of Table 1, we provide a more detailed comparison with prior-art methods in terms of different techniques in Table 6.

Table 6: Comparison with prior-art methods in terms of different techniques.

| Method | Protocol Optimization | Network Optimization | | |
|---|---|---|---|---|
| | | Convolution | Truncation | ReLU |
| [43, 42, 39, 8, 38, 29, 41] | ReLU, Trunc, Conv. | ✗ | ✗ | ✗ |
| [32, 7, 19, 40, 37, 6, 27, 26] | ✗ | ✗ | ✗ | ReLU count/sensitivity-aware NAS |
| [48, 25] | Conv. (Online to Offline) | ✗ | ✗ | ReLU count-aware NAS |
| [17] | Conv. (Winograd-based) | Factorized Point-wise Conv. | ✗ | ✗ |
| [36] | ✗ | Channel Reduction | Channel Reduction | Channel Reduction |
| CoPriv (ours) | Conv. (Winograd-based) | Re-paramerization | Re-paramerization | Communication-aware NAS |

# G  Comparison of Convolution Protocol with HE-Based Method

We also compare CoPriv with SOTA homomorphic encryption (HE) based method, Cheetah [24], which uses HE-based protocol for convolution instead of oblivious transfer (OT) in CrypTFlow2. Cheetah achieves lower communication compared to CrypTFlow2 at the cost of more computation

overhead for both the server and client. Here, we compare their inference latency. Hence, we believe Cheetah and CrypTFlow2/CoPriv have different applicable scenarios. For example, for less performant clients, Cheetah may not be applicable, while when the bandwidth is low, CrypTFlow2/CoPriv may not be the best choice. When comparing CoPriv with Cheetah, we observe a similar trend. As shown in Table 7, we find while CoPriv always outperforms CrypTFlow2, for high bandwidth, CoPriv achieves lower latency compared to Cheetah while for low bandwidth, Cheetah incurs lower latency.

Table 7: Convolution latency (s) comparison with Cheetah for different blocks (expand ratio is set to 6) and bandwidths.

| Bandwidth | 384 MBps [24] | 44 MBps [24] | 9 MBps [47, 39] |
|---|---|---|---|
| Dimension: $(4 \times 4 \times 64)$ | | | |
| Baseline OT (CrypTFlow2) | 2.14 | 3.55 | 16.2 |
| HE (Cheetah) | 1.35 | 1.54 | 2.99 |
| CoPriv w/ Winograd (ours) | 0.21 | 0.64 | 3.00 |
| Dimension: $(8 \times 8 \times 32)$ | | | |
| Baseline OT (CrypTFlow2) | 1.29 | 3.94 | 16.1 |
| HE (Cheetah) | 0.72 | 0.73 | 1.53 |
| CoPriv w/ Winograd (ours) | 0.17 | 0.45 | 2.16 |

