# OpenReview forum: "CoPriv: Network/Protocol Co-Optimization for Communication-Efficient Private Inference"
_NeurIPS.cc/2023/Conference — NeurIPS 2023 poster_

### Official Review · Reviewer_P7t8 · 2023-06-11

**Soundness:** 2 fair
**Presentation:** 3 good
**Contribution:** 2 fair
**Rating:** 6
**Confidence:** 4

**Summary:**

This paper presents a framework that simultaneously optimizes the 2PC inference protocol and the neural network architecture to achieve a significant reduction in communication. The framework outperforms state-of-the-art (SOTA) approaches by achieving communication reduction.

**Strengths:**

1. The paper is well-written and easy to understand
2. The paper highlights the current scenario where nonlinearity no longer dominates the communication overhead.

**Weaknesses:**

1. It would be beneficial to include more ablation experiments to demonstrate the individual contributions of the ReLU pruning and re-parameterization approaches.

2. Providing the percentage or number of multiplication reduction resulting from the Winograd algorithm would provide additional insights into the overall communication reduction.

3. Please ensure that the legends are appropriately positioned within the figures, specifically those that cover certain data points.

4. Empirically comparing the proposed ReLU pruning method with prior methods would provide a deeper understanding of its effectiveness.

5. The most recent work on the ReLU reduction is as follows, the author could consider including it in the introduction:
"S. Kundu, et al., Making Models Shallow Again: Jointly Learning to Reduce Non-Linearity and Depth for Latency-Efficient Private Inference, CVPRW 2023."

**Questions:**

Please refer to the weaknesses section.

**Limitations:**

Please refer to the weaknesses section.

---

> ### Author Rebuttal · Authors · 2023-08-02
>
> We sincerely thank Reviewer P7t8 for your thoughtful feedback!
>
> ---
> **Q1:** Include more ablation experiments to demonstrate the individual contributions of ReLU pruning and re-parameterization.
>
> **A1:** Thanks for your suggestion! In Section 5.5, we perform an ablation study by adding our proposed techniques step by step. And we also add more ablation experiments with 60% and 30% ReLU remained shown in the table below.
> For 30% ReLU, CoPriv achieves 1.4$\times$ online communication reduction after pruning, 3.2$\times$ and 3.6$\times$ online/total communication reduction after re-parameterization compared with baseline MobileNetV2. All of these results indicate that our proposed optimizations are indispensable for improving the communication efficiency.
>
> Model (60% ReLU) | Online Comm. (GB) | Total Comm. (GB)
> ------ | ------ | ------
> Baseline MobileNetV2  | 0.82 | 8.00
> +Pruning  | 0.64 | 7.81
> +Re-parameterization  | 0.43 |  5.62
>
> Model (30% ReLU) | Online Comm. (GB) | Total Comm. (GB)
> ------ | ------ | ------
> Baseline MobileNetV2  | 0.82 | 8.00
> +Pruning  | 0.58 | 7.76
> +Re-parameterization  | 0.26 |  2.21
>
> ---
>
> **Q2:** Providing the percentage or number of multiplication reduction resulting from the Winograd algorithm would provide additional insights into the overall communication reduction.
>
> **A2:** For ResNet-18 and 32 with regular convolutions, Winograd algorithm reduces the number of multiplications by 2.25$\times$ with $F(2\times2, 3\times3)$ transformation theoretically and empirically (shown in our micro-benchmark in Section 5.2), resulting $\sim 2.1\times$ communication reduction;
> And for our proposed CoPriv, we compute the number of multiplications of original MobileNetV2 and CoPriv with 40% ReLUs remained which is shown in **Figure 1 in the rebuttal PDF**. From this, we observe that the number of multiplication is significantly reduced to 68% with our Winograd algorithm, resulting a lower communication.
>
> ---
>
> **Q3:** Ensure that the legends are appropriately positioned within the figures.
>
> **A3:** Thanks for your kind suggestion on our figures!  We will carefully improve our writing and figures in the revised version.
>
> ---
>
> **Q4:** Empirically comparing the proposed ReLU pruning method with prior methods would provide a deeper understanding of its effectiveness.
>
> **A4:** The key insight of our ReLU pruning is 1) to directly use communication to guide the pruning instead of proxy metrics, e.g., ReLU count, and 2) to enforce two ReLUs within the same block to share $\alpha$ so that the two ReLUs can be removed simultaneously to enable re-parameterization and Winograd-based optimization. In contrast, SENet/DeepReDuce/SNL just focus on reducing ReLU count and thus, can barely reduce the total communication. So, our focus is mainly on the optimization pattern rather than the pruning algorithm itself, and note that other pruning method can be easily plugged into our proposed framework.
>
> In this paper, we have compared CoPriv with ReLU-optimized methods including SNL, SENet and DeepReDuce in Section 5.3 and 5.4. Among them, SNL and SENet propose NAS (SENet further uses pruning sensitivity to analyze ReLU importance) to prune ReLUs while DeepReDuce manually decides which ReLUs to be removed.
> One big difference between our pruning method and prior methods is the $L_{comm}$, which means ReLUs in different layers are not equivalent, and makes our method communication-aware rather than ReLU count-aware. We discover that $L_{comm}$ helps us better trade-off the efficiency and accuracy.
>
> To show the importance of introducing $L_{comm}$, we compare our pruning method w/ $L_{comm}$ and pruning methods of SNL/SENet w/o $L_{comm}$ in the **Figure 2 in the rebuttal PDF**. As we can see, $L_{comm}$ helps to focus the pruning on the later inverted residual blocks, which incur more communication overhead, and our method effectively penalizes the importance of the costly blocks (e.g., block #16 and #18), achieving significantly lower communication. By adjusting $L_{comm}$, CoPriv has a strong ability to better trade-off the efficiency and accuracy.
>
> ---
>
> **Q5:** The author could consider including the most recent work (CVPRW 2023) on the ReLU reduction in the introduction.
>
> **A5:** Thanks for pointing out the valuable work. We will include this paper in our revised version. We also make the following comparison:
>
> 1) Similarity: the two papers both reduce the ReLUs and the network depth.
>
> 2) Motivations are different: the CVPRW paper still regards ReLU as the main latency bottleneck and removes convolution layers to reduce computation. In contrast, we fuse neighboring convolution layers to reduce truncations and better leverage our Winograd-based optimization for communication reduction. The difference in motivation leads to different criterion when selecting convolutions to remove. The CVPRW paper selects convolutions based on ReLU sensitivity while we consider both accuracy and communication cost.
>
> 3) Methods are different: the CVPRW paper first determine which convolutions to remove and then, use the gated branching mechanism to train new convolutions. In contrast, we simultaneously train the architecture parameters $\alpha$ with the model weights and the re-parameterization is conducted **post training**, enabling us to better leverage the benefits of over-parameterization as shown in the RepVGG paper (RepVGG: Making VGG-style ConvNets Great Again).
>
> As shown in the table below, our method achieves both better accuracy and lower communication compared to the CVPRW paper.
>
> Method | Top-1 Acc. (%) | Online Comm. (GB) | Total Comm. (GB)
> ------  | ------ | ------ | ------
> CVPRW 2023 | 69.10 | 0.81 | 46.5
> CoPriv (ours) | 70.58 | 0.43 | 5.14

---

### Official Review · Reviewer_AJR8 · 2023-06-27

**Soundness:** 4 excellent
**Presentation:** 4 excellent
**Contribution:** 3 good
**Rating:** 7
**Confidence:** 2

**Summary:**

The paper presents optimizations to secure two-party computation of convolutional network inference. There are optimizations for both linear and non-linear layers, resulting in an overall single-digit factor improvement.


**Strengths:**

The optimizations look interesting and are underlined well with benchmarks. I particularly appreciate the trade-off visualization in Figure 8.


**Weaknesses:**

Table 1 doesn't make sense to me. I don't think there is merit in pointing out that prior work hasn't optimized certain aspects because it might be that the work is efficient without extra effort. If anything, the table should contain numerical speed-ups.

Using the acronym ASS for arithmetic secret sharing might put off readers as it's identical to a vulgar word.

**Questions:**

n/a

---

> ### Author Rebuttal · Authors · 2023-08-02
>
> We sincerely thank Reviewer AJR8 for your thoughtful feedback!
>
> ---
>
> **Q1:** Table 1 should contain numerical speed-ups.
>
> **A1**: We thank the reviewer for the valuable feedback. As shown in Figure 1 in the paper, operations like ReLU, truncation, and convolution, are major contributors to the online or total communication. To reduce their communication, we hope to emphasize the importance of both protocol and network optimization, and all the components of the neural network should be fully considered, and we also compare our CoPriv with existing methods. Hence, we have a qualitative comparison in Table 1 and leave quantitative comparison in experimental results.
>
> We do agree simple qualitative comparison may not be very useful. We augment the table in **Table 2 in the rebuttal PDF** following your suggestion to include numerical speed-ups in the table. We will think of better ways to make the comparisons as well.
>
> ---
>
> **Q2:** Using the acronym ASS for arithmetic secret sharing might put off readers as it's identical to a vulgar word.
>
> **A2:** Thanks for your advice! We agree with your opinion, and we will consider a more appropriate acronym for arithmetic secret sharing like SS or ArSS in our revised version.

---

> > ### Comment · Reviewer_AJR8 · 2023-08-16
> >
> > Q1: I'm not arguing against the benefit of optimization, I'm just saying that I cannot think of an objective definition of what constitutes optimized or not. The improvement figures are much appreciated, but I still don't see the point of having ticks for optimization.

---

> > > ### Author Response · Authors · 2023-08-16
> > >
> > > Thanks for your valuable suggestion and comment!
> > > We admit that simply using ticks for optimization is not appropriate.
> > > Therefore, we modify this table as shown below to include more details to make the comparison more clear for understanding.
> > > The below table compares our CoPriv with prior works in terms of the optimized algorithms as well as the used techniques.
> > >
> > > |  Method   | Protocol Opt  |  Network Opt: Conv  |  Network Opt: Trunc | Network Opt: ReLU
> > > |  ----  | ----   | --- | --- | --- |
> > > | [33, 32, 30, 6, 29] | ReLU, Trunc, Conv | - | -| - |
> > > | [25, 5, 16, 31, 28, 4] | - | - | - | ReLU count-aware/sensitivity-aware NAS |
> > > | [37, 20]  | Conv (Online Comm. to Offline Comm.) | - | - | ReLU count-aware NAS |
> > > | [27] | - | Channel Reduction | Channel Reduction | Channel Reduction |
> > > | CoPriv (ours) | Conv (Winograd-based Protocol) | Re-paramerization | Re-paramerization | Communication-aware NAS |
> > >
> > > The descriptions and comparisons of the mentioned works are included in our Related Works in Appendix A.

---

### Official Review · Reviewer_vLvc · 2023-07-03

**Soundness:** 3 good
**Presentation:** 3 good
**Contribution:** 3 good
**Rating:** 6
**Confidence:** 4

**Summary:**

The paper introduces CoPriv, a framework that optimizes the 2-party computation (2PC) inference protocol and the deep neural network (DNN) architecture to reduce communication overhead. CoPriv features a new 2PC protocol for convolution based on Winograd transformation and develops DNN-aware optimization to reduce inference communication.

**Strengths:**

The authors highlight a significant point that pruning ReLU may no longer be the most effective method for reducing computational and communication costs in private inference. This is due to the increasing prominence of linear and truncation operations. This insight is of considerable importance to the community.

**Weaknesses:**

1. Figure 1 could benefit from more specific details. The left figure should include specific numbers for each portion and the amount of communication cost reduction achieved by each technique. It's also unclear which private inference method is used in the left figure.

2. The paper does not clearly explain why the DNN-aware adaptive convolution protocol can reduce communication costs. As a major contribution, it would be beneficial if the authors could provide a detailed explanation of how the selection of protocol initializer impacts communication costs and the criteria for selecting the initializer.

3. The novelty of the proposed ReLU pruning method is not clear unless the authors can explain how L_{comm} affects the training results. It would be valuable to compare the proposed ReLU pruning method with DeepReduce/SNL/SENet in terms of which ReLUs remain in the network and final accuracy.

4. It would be beneficial to differentiate the proposed re-parameterization method from the following work:
"Making Models Shallow Again: Jointly Learning to Reduce Non-Linearity and Depth for Latency-Efficient Private Inference" (https://openaccess.thecvf.com/content/CVPR2023W/ECV/papers/Kundu_Making_Models_Shallow_Again_Jointly_Learning_To_Reduce_Non-Linearity_and_CVPRW_2023_paper.pdf).

5. From Figure 9, it appears that the authors do not apply ReLU pruning and re-parameterization to each block. The criteria for deciding which blocks are suitable for ReLU pruning and re-parameterization are not clear.

6. In Table 4, the accuracy remains the same for MobieNetV2 with and without pruning+re-parameterization. Since pruning and re-parameterization typically lead to accuracy degradation, could the authors provide an explanation for these results?

7. Given that Cheetah outperforms CrypTFlow2, it would be more valuable and informative to compare the proposed protocol optimization method with Cheetah rather than CrypTFlow2. This comparison could provide a more accurate assessment of the proposed method's performance relative to the current state-of-the-art.

**Questions:**

Please refer to the weakness section.

**Limitations:**

While the authors acknowledge the limitation of Winograd convolution, stating that it can only be applied to 3x3 depth-wise convolution, they do not discuss the limitations of their entire work.

---

> ### Author Rebuttal · Authors · 2023-08-03
>
> We sincerely thank Reviewer vLvc for your thoughtful feedback!
>
> ---
>
> **Q1:** Figure 1 could benefit from more details.
>
> **A1:** Thanks for your advice! We will improve this figure carefully to include more details.
> For ReLU, [22] represents Gazelle which uses garbled circuit (GC), [33] represents CrypTFlow2 which uses IKNP OT, and [19] represents Cheetah which uses VOLE OT. We get these numbers directly from each paper.
>
> ---
>
> **Q2:** Not clearly explain the DNN-aware adaptive convolution protocol.
>
> **A2:** As introduced in Section 4.1, in the Winograd-based convolution protocol, with tile aggregation, the server and client need to jointly run the OT-based matrix multiplication protocol for $(m+r-1)^2$ times. For each OT-based matrix multiplication, the server and the client hold the weight and activation of shape $(K, C)$ and $(C, T)$, respectively, where $K$, $C$, and $T$ denote the number of output channels, input channels, and # tiles, respectively, and are impacted by the DNN architecture. We observe the cost of the OT-based matrix multiplication depends on the OT initializer. More specifically, when the server initializes the OT, the round of communication and the communication of each round are $O(CK)$ and $O(\lambda + T)$, respectively (the complexity is derived based on CrypTFlow2). In contrast, when the client initializes the OT, the round and communication of each round become $O(CT)$ and $O(\lambda + K)$, respectively. As $K$, $C$, and $T$ are known before inference, our DNN-aware adaptive protocol will select the optimal OT initializer for each network and each layer to minimize the communication cost.
>
> ---
>
> **Q3:** How $L_{comm}$ affects training.
>
> **A3:** The key insight of our ReLU pruning is 1) to directly use communication to guide the pruning instead of proxy metrics, e.g., ReLU count, and 2) to enforce two ReLUs within the same block to share $\alpha$ so that they can be removed simultaneously to enable re-parameterization and Winograd-based optimization. In contrast, SENet/DeepReDuce/SNL just focus on reducing ReLU count and thus, can barely reduce the total communication.
>
> To show the importance of introducing $L_{comm}$, we compare our pruning method w/ $L_{comm}$ and pruning methods of SNL/SENet w/o $L_{comm}$ in **Figure 2 in the rebuttal PDF**. As we see, $L_{comm}$ helps to focus the pruning on the later layers, which incur more communication cost, and penalizes the costly blocks (e.g., block #16/#18). In contrast, SENet focuses on pruning early layers with more ReLU counts.
>
> ---
>
> **Q4:** Differentiate the proposed re-parameterization from CVPRW 2023.
>
> **A4**: Thanks for pointing out the valuable work. We will cite this paper in our revised version. We also make the following comparison:
>
> 1) Similarity: the two papers both reduce the ReLUs and the network depth.
>
> 2) Different motivations: the CVPRW paper still regards ReLU as the main latency bottleneck and removes convolution layers to reduce computation. In contrast, we fuse neighboring convolution layers to better leverage our Winograd-based optimization for communication reduction of all operators. The difference in motivation leads to different criterion when selecting convolutions to remove. The CVPRW paper selects convolutions based on ReLU sensitivity while we consider both accuracy and communication cost.
>
> 3) Different methods: the CVPRW paper first determines which convolutions to remove based on ReLU sensitivity and then, uses the gated branching method for training. In contrast, we simultaneously train the architecture parameters with the model weights, and the re-parameterization is conducted **post training**, enabling us to leverage the benefits of over-parameterization shown in RepVGG (RepVGG: Making VGG-style ConvNets Great Again).
>
> As shown in **Table 1 in the rebuttal PDF**, CoPriv achieves both better accuracy and lower communication compared to the CVPRW paper.
>
> ---
>
> **Q5:** The criteria for ReLU pruning and re-param is not clear.
>
> **A5**: We prune ReLUs based on the architecture parameter $\alpha$ of each block. $\alpha$ is trained jointly with model parameters and considers both accuracy and communication cost. During searching, we gradually fix small $\alpha$ to 0 until the required communication is achieved. After ReLU pruning, there are only three sequential convolutions in the inverted residual block. Then, we further re-parameterize the block into a single convolution.
>
> ---
>
> **Q6:** Explain the accuracy degradation of pruning and re-param.
>
> **A6:** MobileNetV2 with pruning degradates the accuracy compared with original MobileNetV2, but here we directly report the accuracy of MobileNetV2 after pruning. In contrast, re-parameterization will not hurt the accuracy because it **equivalently** merges the convolutions within an inverted residual block into a single convolution **post training**. We show the detail of re-parameterization in Appendix C.
>
> ---
>
> **Q7:** Compare the protocol optimization with Cheetah.
>
> **A7**: Thanks for the valuable advice. Cheetah uses HE-based protocol for convolution instead of OT in CrypTFlow2. Cheetah achieves lower communication compared to CrypTFlow2 at the cost of more computation overhead for both the server and client. Hence, we believe Cheetah and CrypTFlow2/CoPriv have different applicable scenarios. For example, for less performant clients, Cheetah may not be applicable while when the bandwidth is low, CrypTFlow2/CoPriv may not be the best choice.
>
> When comparing with Cheetah, we observe a similar trend. As in **Table 3 in the rebuttal PDF**, we find while CoPriv always outperforms CrypTFlow2, for high bandwidth, CoPriv achieves lower latency compared to Cheetah while for low bandwidth, Cheetah incurs lower latency. Hence, in our paper, we focus on comparison with CrypTFlow2 and other OT-based baselines.
>
> **Limitation:** There is still an efficiency gap between ciphertext and plaintext inference, and we leave more valuable work as our future work.

---

> > ### Comment · Reviewer_vLvc · 2023-08-16
> > **Thanks!**
> >
> > Thank you to the authors for the clarifications. My concerns are now addressed, and I have accordingly raised my score.

---

### Official Review · Reviewer_yVeD · 2023-07-06

**Soundness:** 3 good
**Presentation:** 3 good
**Contribution:** 3 good
**Rating:** 7
**Confidence:** 4

**Summary:**

This paper presents CoPriv that jointly optimizes 2PC protocols and DNN architectures. It argues that SOTA 2PC protocols mainly focus on minimizing ReLU-based metric, which no longer contributes to the majority of communication. It proposes a new protocol for convolution with Winograd transformation and proposes a series of other DNN-aware optimizations. It shows communication reduction compared to SOTA protocols, CrypTFlow2, and other network optimization methods.

**Strengths:**

(1) The observations in the motivation section are informative, well written and supported.
(2) The Winograd based transformation with tile aggregation is well motivated and well illustrated (Figure 4).
(3) The adaptive convolutional protocol is novel, insightful and effective.
(4) The ablation study which adds optimizations step by step is conclusive and well-conducted.
(5) The proposed methods consistently achieve communication reduction compared to SOTA.

**Weaknesses:**

The implication of end-to-end speedup is not well studied.

**Questions:**

The paper shows a consistent communication reduction, which the reviewer appreciated. However, it would be better if the author can provide some analysis on the end-to-end speedup using common setups.

**Limitations:**

The paper only analyzes the communication volume, where the effect on the communication time, inference speedup has not been well discussed. It can be the case that with higher bandwidth, this communication volume reduction will not be significant. However, the reviewer appreciates the communication volume reduction alone, and would not consider this as a major limitation.

---

> ### Author Rebuttal · Authors · 2023-08-03
>
> We sincerely thank Reviewer yVeD for your thoughtful feedback!
>
> ---
>
> **Q1:** The implication of end-to-end speedup is not well studied.
>
> **A1:** Thanks for the meaningful suggestion on the inference speedup! In our experiments, we compare the inference latency of MobileNetV3 (with different capacities), ReLU-optimized networks (including SNL and SENet), and SOTA pruning methods (including uniform pruning and MetaPruning) on the ImageNet dataset with a widely used LAN communication setting (i.e., 377 MBps bandwidth and 0.3ms echo latency) following CrypTFlow2.
> As shown in Figure 8(c), on the ImageNet dataset, 1) our proposed CoPriv outperforms SNL with 6% higher accuracy and 1.4$\times$ latency reduction, and 2) CoPriv outperforms SENet with 2.8% higher accuracy and 2.2$\times$ latency reduction. These results demonstrate a consistent accuracy improvement with latency reduction.
>
> When the communication bandwidth becomes lower, e.g., in the wireless setting, we expect our latency reduction can be further improved.
> In brief, we do agree with your valuable suggestions on the inference speedup, and we will add more results about the speedup in the wireless setting with a lower bandwidth in the revised version.

---

> > ### Comment · Reviewer_yVeD · 2023-08-16
> >
> > Thanks a lot for the great response. My concerns are addressed. I will keep my positive rating. Please consider accepting it.

---

### Author Rebuttal · Authors · 2023-08-09

We sincerely thank all the reviewers for the thoughtful feedback and helpful comments!

**Rebuttal One-page PDF:** we attach the one-page PDF here to include the figures and tables mentioned in the following responses. We give a brief introduction of these figures and tables below to provide convenience for quick review for the reviewers:

* Figure 1: Comparison of the number of multiplications in each block between original MobileNetV2 and CoPriv with Winograd transformation (based on the comments of Reviewer P7t8).
* Figure 2: Comparison of different pruning method and the influence of $L_{comm}$ during the search in each block (based on the comments of Reviewer vLvc and P7t8).
* Table 1: Accuracy and communication comparison between the CVPRW paper and our CoPriv (based on the comments of Reviewer vLvc and P7t8).
* Table 2: Comparison with prior-art methods with qualitative optimization level (based on the comments of Reviewer AJR8).
* Table 3: Latency comparison with Cheetah for different blocks and bandwidths (based on the comments of Reviewer vLvc).

---

### Decision · Program_Chairs · 2023-09-21

**Decision:**

Accept (poster)

**Comment:**

This work proposes a method that jointly optimizes 2PC protocols and DNN architectures. It argues that other linear and non-linear operations are also important in addition to ReLU which was the focus of previous works.

The final ratings are unanimously accept (6, 6, 7, 7). Reviewers recognize the novelty and clarity of the paper as well as the improvements over STOA.

The authors addressed most of the concerns in the rebuttal. Please add the additional discussions and experiments in the final version.